# `MMD-NSL`: Mixed Multinomial Distribution-based Neuro-Symbolic Learning

## Abstract

Neuro-symbolic learning (NSL) aims to integrate neural networks with symbolic reasoning approaches to enhance the interpretability of machine learning models. Existing methods mostly focus on the long dependency problem of symbolic learning. The important challenge of complex categorization is largely overlooked. To bridge this gap, we propose the Mixed Multinomial Distribution-based NSL (`MMD-NSL`) framework. It seamlessly integrates the handling of long dependency chains and complex semantic categorization within Knowledge Graphs (KGs). By introducing a continuous Mixed Multinomial Logic Semantic Distribution, we extend traditional Markov Logic Networks (MLN) to incorporate context-aware semantic embeddings. Our theoretical innovations, including a bijective mapping between MLNs and continuous multinomial distributions, enable the capture of intricate dependencies and varied contexts crucial for NSL tasks. The framework leverages a bilevel optimization strategy, where a transformer-based upper level dynamically learns mixing coefficients akin to attention mechanisms, while the lower level optimizes rule weights for learning both context and rule patterns. Extensive experiments on the DWIE benchmarking datasets demonstrate significant advantages of `MMD-NSL` over four state-of-the-art approaches. It achieves 10.47% higher F1-scores on average than the best-performing baseline across 23 sub-datasets. It advances continuous probabilistic models for neuro-symbolic reasoning and complex relational tasks.

## 1 Introduction

Neuro-symbolic learning (NSL) d'Avila Garcez et al. (2012); Lamb et al. (2020); Besold et al. (2021) represents a promising frontier in artificial intelligence, aiming to integrate the robustness of neural networks LeCun et al. (2015); Schmidhuber (2015) with the interpretability and formal reasoning capabilities of symbolic approaches Newell & Simon (1956); McCarthy (1960). By combining the two, NSL seeks to address the limitations of traditional neural models, which often struggle with transparency and the ability to handle structured, rule-based reasoning tasks.

A key objective of NSL is to enhance question-answering (QA) systems by enabling logical reasoning over complex logic steps. These reasoning processes resemble the "chain of thought" prompt approach Wei et al. (2022); Kojima et al. (2022) utilized by large language models (LLMs) Brown et al. (2020); OpenAI (2023) to reason through problems step by step. As shown in Fig 1, NSL leverages accumulated historical data to refine the reasoning process, moving beyond traditional symbolic learning approaches that rely on directly learning hard rules. Instead, NSL emphasizes the discovery of higher-level patterns in a differentiable framework, allowing for robust handling of logically similar questions.

Consider, for example in Fig 1, two related QA tasks: *Under what circumstances do many doctors appear at the schoolyard?* and *Under what circumstances do police choose to patrol at intersections?* Both questions require reasoning about specific categories of people appearing in particular locations. For the second question, reasoning paths might involve long dependency chains such as *Car accidents → Traffic jams → Conflicts among people → Police presence* or *Important events → Crime prevention → Police presence*. These examples highlight the complexity of reasoning over such logical chains, as well as the importance of uncovering high-level generalizable patterns that can handle these dependencies.

How to improve answer for **similar** questions
with accumulated historical data for a Q&A task language model?
• **Q1:** Under what circumstances do many doctors appear at schoolyard?
• **Q2:** Under what circumstances do police choose to patrol at intersections?
• **A1 for Q2:** Car accidents in intersections -> Traffic jams -> Conflict among people ->
Police presence
• **A2 for Q2 :** Important event in intersections -> Crime prevention -> Police presence

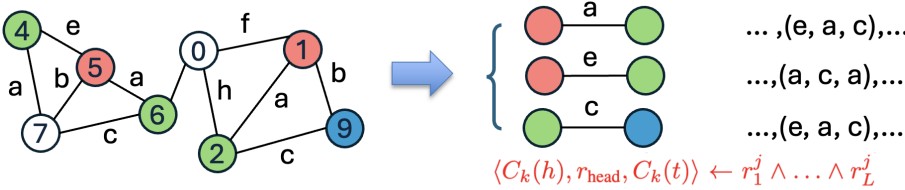

**Mapping into Mixed Multinomial Logic Semantic Distribution**

Figure 1: The figure illustrates an overview of the proposed framework. At the top, it depicts a question-answering learning scenario that not only requires capturing long dependency chains but also involves learning "similar" context categorizations. These categorizations, marked by red and yellow text highlights, emphasize examples of people-location relationships with similar contextual patterns. The bottom section demonstrates how the above scenario can be seen as an equivalent path search problem for a realtion query with its NER-pair context. The bottom-left portion represents the KG structure, where nodes are color-coded to indicate different NER types, visualizing the node-based semantic space. On the bottom-right, the diagram transitions to a categorized NER-based space. In this space, the path search problem is conceptualized as drawing path samples from a mixed multinomial logic distribution, integrating both logical and contextual semantics.

Two significant challenges arise in this context. First, the reasoning process involves navigating a vast search space of possible logical paths, making it impractical to enumerate all potential scenarios. Second, the system must operate in a continuous semantic space that can effectively model and categorize diverse entity types, such as "doctors" and "police," while capturing their contextual relationships. Addressing these challenges requires an approach that combines symbolic reasoning with probabilistic and neural techniques.

This reasoning task can be reframed as a Knowledge Graph (KG) problem as shown in Fig 1, where answering a question corresponds to finding equivalent logical paths within the graph. To reduce the complexity of the search space, nodes in the KG (e.g., "doctors" or "schoolyard") are mapped into a Named Entity Recognition (NER) embedding space. This transformation allows the model to capture high-level semantic relationships, enabling reasoning over multi-entity combinations and providing a structured representation for logical reasoning.

In summary, addressing the challenges in NSL requires two critical capabilities: (1) managing **long dependency chains** and (2) handling **complex semantic categorization** across diverse question types involving NER combinations. While significant advances have been made in addressing long dependency chains, existing approaches often overlook the complexity of semantic categorization. This limitation restricts the ability of current NSL models to tackle real-world scenarios, where entities and relationships often demonstrate diverse and context-sensitive semantics.

To overcome this gap, we propose the Mixed Multinomial Distribution-based NSL (`MMD-NSL`) framework. `MMD-NSL` generalizes the modeling of logical rules and NER-based contexts into a unified probabilistic framework, leveraging mixed multinomial logic semantic distributions. By incorporating context-aware probabilistic reasoning, this approach bridges the gap between rule-based reasoning and context-dependent categorization, enhancing the scalability and interpretability of NSL models. `MMD-NSL` accommodates both complex categorical contexts and long depen-

dency chains, allowing it to address the limitations of prior methods by modeling structured relations through context-specific paths.

The flexibility of `MMD-NSL` is further enhanced by its bilevel optimization strategy. At the upper level, transformer-based architectures dynamically learn the continuous context logic semantic distribution as mixing coefficients, similar to attention mechanisms, to capture complex categorization. At the lower level, rule weights are optimized to refine the modeling of long dependency chains and symbolic patterns.

From a theoretical perspective, we demonstrate that `MMD-NSL` establishes a bijective mapping between Markov Logic Networks (MLN) and continuous multinomial distributions. This mapping extends the capabilities of MLNs to incorporate categorical contexts through continuous variables, unifying rule-based reasoning and semantic categorization within a probabilistic framework. By treating logical rules and NER combinations as latent variables, `MMD-NSL` relaxes discrete contexts into continuous semantic spaces, enabling flexible modeling and efficient optimization.

To validate `MMD-NSL`, we conducted extensive experiments on the DWIE benchmark, which evaluates the ability to reason over complex semantic relationships across 23 diverse sub-datasets. Our results show that `MMD-NSL` achieves a substantial 10.47% improvement in F1-score over four state-of-the-art NSL models, demonstrating its superiority in handling both long dependency chains and semantic categorization.

## 2 RELATED WORKS

To model logic rules as latent variables in a softened form is a standard approach in many probabilistic NSL frameworks Bach et al. (2017); Dong et al. (2019); Manhaeve et al. (2018); Trouillon et al. (2016) including softened logic operator Maene & De Raedt (2024) or softened reasoning architecture Marra et al. (2021). It introduces flexibility by relaxing strict logical constraints, allowing the model to handle uncertainties and partial truths.

A significant line of research focuses on Markov Logic Network (MLN)-based NSL, which are a type of graphical probabilistic model. Although symbolic logic learning effectively captures complex long dependencies, its deterministic nature makes it less suited for uncertain scenarios. To address this, MLNs Richardson & Domingos (2006) integrate logic with probabilistic graphical models by assigning weights to logical formulas, allowing for softened rule structures. Enhancements in MLN-based models include Lifted Inference Braz et al. (2005); Singla & Domingos (2008); Sourek et al. (2018); Wu et al. (2020), Lazy Inference Singla & Domingos (2006); Poon et al. (2008), and Coarse-to-Fine Inference Kiddon & Domingos (2011), all aimed at improving efficiency and scalability. Notable developments in this domain include Logic Tensor Networks (LTN) Donadello et al. (2017), which relax logical operators into fuzzy logic using t-norm theory, and Neural Markov Logic Networks (NMLN) Marra et al. (2019) and Relational Neural Machines Marra et al. (2020), which employ neural networks to model real-valued differentiable functions in probabilistic logic. In KG context, semantic logic learning focus on logic chain structure decomposition Cheng et al. (2022) and composition Cheng et al. (2023). All the above works can be classified under the chain structure-based NSL direction.

On the other hand, treating logic chains as samples drawn from a rule distribution represents a distinct generative direction, often based on Multinomial Distributions (MD). This NSL direction leverages multinomial distributions to effectively model rules within neuro-symbolic frameworks. For example, Qu et al. (2020) proposed modeling the rule body in the NSL task as a sample drawn from a multinomial distribution, which can be parameterized by sequence models like Transformers. Consequently, many Transformer-based NSL models have gained significant attention in the literature Ru et al. (2021); Xing et al. (2023), where logic rules are learned as latent variables within a multinomial distribution framework. Even though chain structure-based NSL direction have started to concern context information of neighbors in subgraphs Han et al. (2023), distribution-based generative NSL approaches do not incorporate mixtures of distributions to account for category contexts. This gap highlights the need to extend these frameworks by integrating category contexts to enhance the semantic richness and interpretability of NSL models. The proposed `MMD-NSL` framework bridges this gap.

## 3 METHODOLOGY

### 3.1 OVERVIEW

Our approach extends traditional NSL tasks by incorporating complex category contexts into the probabilistic framework. In the context of a KG, NSL tasks typically consider long dependencies expressed in the form $r_{\text{head}} \leftarrow r_1^j \wedge \ldots \wedge r_L^j$, where $j$ represents the $j$-th unique rule body for the current $r_{\text{head}}$, and $L$ is the length of the relation along a rule body path. We generalize this to tasks formulated as $\langle C_k(h), r_{\text{head}}, C_k(t) \rangle \leftarrow r_1^j \wedge \ldots \wedge r_L^j$, where $C_k(\cdot)$ denotes the $k$-th NER-pair type of the head-tail nodes, and $h$ and $t$ are the head and tail nodes of the relation $r_{\text{head}}$. The rule bodies $r_1^j \wedge \ldots \wedge r_L^j$ correspond to equivalent paths under the same NER type combination as the rule head $r_{\text{head}}$ within the graph $\mathcal{G}$.

The motivation behind our approach is to treat logical rules and NER combinations as latent variables and to map them into continuous spaces. Initially, we relax the discrete rule and context variables using semantic mappings. These relaxed, continuous variables are then paired with rule weights and context weights to form two distributions: a continuous rule distribution to capture long dependencies, and a continuous context distribution to account for complex categorical contexts. This unified framework enables simultaneous modeling of both rule-based logical structures and the category-specific contexts present in KGs. Under this structure, fuzzy logic can be seen as a specific instance of continuous logic variables, and MLN as a form of continuous logic distribution. Existing approaches based on multinomial distributions are also naturally subsumed as a special case where context complexity is a single context scenario. This unified variable and distribution framework allows for a comprehensive treatment of logical rules and contexts in NSL, bridging the gap between rule-based reasoning and categorical context modeling within probabilistic structures.

### 3.2 MIXED MULTINOMIAL LOGIC SEMANTIC DISTRIBUTION

Let $\boldsymbol{z} = \{z_j\}_{j=1}^J$ be a set of latent variables corresponding to $J$ unique rules for a $r_{\text{head}}$ defined over a KG $\mathcal{G}$. Similarly, let $\boldsymbol{C} = \{C_k\}_{k=1}^K$ represent a set of latent context variables, each corresponding to distinct category combinations (e.g., $K$ different combinations of Named Entity Recognition (NER) types). We denote $z_j \mid C_k$ as the $j$-th rule variable conditioned on the $k$-th context variable, which specifically corresponds to the structured relation $\langle C_k(h), r_{\text{head}}, C_k(t) \rangle \leftarrow r_1^j \wedge \ldots \wedge r_L^j$. Both $z_j$ and $C_k$ are discrete variables, where $z_j = 1$ if the rule $z_j$ holds true in $\mathcal{G}$ and $-1$ otherwise, corresponding to the relation label $y$ in the KG. The context variable $C_k$ represents a discrete combination of all possible NER types.

#### 3.2.1 BACKGROUND

In traditional NSL, various methods such as the pioneering MLN based on probabilistic graphical models, as well as their extensions through fuzzy logic and neural networks, are commonly utilized to model long dependency relationships. These approaches achieve this by introducing softened weights and feature functions, which depend on the underlying mechanism—whether fuzzy operators or neural network-based transformations.

These methods follow a unified foundational formulation, expressed as:

$$P_{\text{MLN}}(\boldsymbol{z}) = \frac{1}{Z} \exp \left( \sum_{j=1}^J w_j f_j(z_j) \right), \tag{1}$$

where $Z$ is a normalization constant known as the partition function, $w_j$ are the weights assigned to each logical formula (feature), and $f_j(z_j)$ are the feature functions, determined by either fuzzy operators or neural network mechanisms, which count the true groundings of the formulas. The summation is over all unique formulas, indexed by $j$ from 1 to $J$.

**Definition 1.** *Logic Semantic Variable*

*The* Logic Semantic Variable *are the continuous variable in $\mathbb{R}$, derived by relaxing the discrete logical variables $z_j$ and contextual variables $C_k$ through the functions $F_{context}(\cdot)$ or $F_{rule}(\cdot)$.*

Following Definition 1, there are two types of Logic Semantic Variable:

**Context Logic Semantic Variable**

$$F_{\text{context}}(C_k) = \mathbf{e}_{C_k}^\top \left( \mathbf{e}_{\text{NER}_{\text{head}}} \odot \mathbf{e}_{\text{NER}_{\text{tail}}} \right), \tag{2}$$

where $\mathbf{e}_{C_k}$ is a embedding vector specific to the $k$-th context combination, capturing the interaction dynamics between the head and tail NER types. $\mathbf{e}_{\text{NER}_{\text{head}}}$ and $\mathbf{e}_{\text{NER}_{\text{tail}}}$ are embedding vectors for the head and tail entities' NER types, respectively. $\odot$ denotes element-wise multiplication.

**Fuzzy Logic Semantic Variable**   When $K = 1$ (i.e., a single context):

$$F_{\text{rule}}(z_j) = \frac{1}{2} y(r_{\text{head}}) \cdot \max_{\mathcal{G}} \prod_{l=1}^{L} \mathbf{e}_{\text{path}}(r_l), \tag{3}$$

When $K > 1$ (i.e., multiple contexts):

$$F_{\text{rule}}(z_j \mid C_k) = \frac{1}{2} y(r_{\text{head}} \mid \mathbf{e}_{C_k}^\top \left( \mathbf{e}_{\text{NER}_{\text{head}}} \odot \mathbf{e}_{\text{NER}_{\text{tail}}} \right)) \cdot \max_{\mathcal{G}} \prod_{l=1}^{L} \mathbf{e}_{\text{path}}(r_l), \tag{4}$$

where $y(r_{\text{head}} \mid C_k) \in \{1, -1\}$ indicates the ground truth label for the relation $r_{\text{head}}$ under context $C_k$, representing whether the relation is true (positive) or false (negative). The terms $\mathbf{e}(r_l) \in [0, 1]$ denote the fuzzy truth values of the relations $r_l$ in the rule body, obtained from the KG embeddings. The product $\prod_{l=1}^{L} \mathbf{e}_{\text{path}}(r_l)$ represents the fuzzy conjunction (t-norm) of the relations along a specific path $z_j$, combining the fuzzy truth values (embedding mapping) of the relations $r_1, \ldots, r_L$. The max operator $\max_{\mathcal{G}}$ represents a fuzzy disjunction (t-conorm) over multiple paths in the graph $\mathcal{G}$, selecting the highest-scoring path to focus on the strongest evidence.

**Definition 2.** *Logic Semantic Distribution*

*The* Logic Semantic Distribution *is the probability distribution $P$ over the continuous variables obtained after applying the Logic Semantic functions $F_{rule}(\cdot)$ and $F_{context}(\cdot)$, combined with the distribution decision weights $\boldsymbol{\theta}$ and $\boldsymbol{\phi}$ (representing learnable context embeddings for $\mathbf{e}_{NER_{head}} \odot \mathbf{e}_{NER_{tail}}$ ).*

Following Definition 2, there are three Logic Semantic Distributions:

**Multinomial Logic Semantic Distribution (for $K = 1$)**

$$P_{\text{MD}} = P(\boldsymbol{\theta}, F_{\text{rule}}(\boldsymbol{z})), \tag{5}$$

where $\boldsymbol{\theta} = \{\theta_j\}_{j=1}^{J}$ is the set of weights associated with each rule $z_j$, determining how the feature functions $F_{\text{rule}}(\boldsymbol{z})$ are weighted in the overall Semantic distribution. Since $K = 1$, the contextual variables $C_k$ are not needed, simplifying the structure to focus on the rule-based logic without considering multiple semantic contexts.

**Context Logic Semantic Distribution**   In this work, we soften the mixing coefficients into a distribution $\boldsymbol{\pi} = \{\pi_k\}_{k=1}^{K}$, which is defined as:

$$\boldsymbol{\pi} = P(\boldsymbol{\phi}, F_{\text{context}}(\boldsymbol{C})), \tag{6}$$

where $\boldsymbol{\phi} = \{\phi_k\}_{k=1}^{K}$ is the set of context decision weights, with each $\phi_k$ determining the influence of context $C_k$ within the distribution.

**Mixed Multinomial Logic Semantic Distribution (for $K > 1$)**

$$P_{\text{MMD}} = P(\boldsymbol{\phi}, F_{\text{context}}(\boldsymbol{C}), \boldsymbol{\theta}, F_{\text{rule}}(\boldsymbol{z} \mid \boldsymbol{C})). \tag{7}$$

**Theorem 1.** *In the case of $K = 1$, assume that there exist $J$ unique rules derived from $N$ samples, and approximate $\log \sigma(z)$ as $\frac{1}{2} z$ via a second-order Taylor expansion, omitting the constant term $-\log(2)$. There exists a fuzzy logic semantic function $F_{rule}(z_j)$, which has the form as in Eq. 3, that establishes a bijective mapping from the MLN-based logic semantic distribution in Eq. 1 to a multinomial distribution-based logic semantic distribution:*

$$P_{MD}(\boldsymbol{z}) = P(\boldsymbol{\theta}, F_{rule}(\boldsymbol{z})) = \frac{1}{Z(\boldsymbol{\theta})} \frac{N!}{\prod_{j=1}^{J} n_j!} \exp\left( \sum_{j=1}^{J} \theta_j \cdot n_j \cdot F_{rule}(z_j) \right), \tag{8}$$

where $n_j$ is the count of rule $z_j$, $\frac{1}{Z(\boldsymbol{\theta})} \frac{N!}{\prod_{j=1}^{J} n_j!}$ corresponds to the normalization factor $\frac{1}{Z}$ in MLN, and the mapping aligns $w_j$ with $\theta_j$, and $f_j(z_j)$ with $n_j \cdot F_{rule}(z_j)$. The $Z(\boldsymbol{\theta})$ is:

$$Z(\boldsymbol{\theta}) = \left( \sum_{j=1}^{J} e^{\theta_j \cdot F_{rule}(z_j)} \right)^N . \tag{9}$$

**Theorem 2.** *Given that the attention weights $\alpha_k$ are computed using the scaled dot-product mechanism:*

$$\alpha_k = \frac{\exp\left( \frac{Q^\top K_k}{\sqrt{d}} \right)}{\sum_{k'=1}^{K} \exp\left( \frac{Q^\top K_{k'}}{\sqrt{d}} \right)}, \tag{10}$$

*where $Q = \mathbf{e}_{NER_{head}} \odot \mathbf{e}_{NER_{tail}}$ is the element-wise product of the head and tail NER embeddings as defined in Eq. 2, and $K_k = \mathbf{e}_{C_k}$ is the context-specific embedding vector for the $k$-th context combination, then under the proper scaling factor $\sqrt{d}$, the attention weights approximate the Context Logic Semantic Distribution $\pi_k$:*

$$\pi_k = \frac{\exp\left( F_{context}(C_k) \right)}{\sum_{k'=1}^{K} \exp\left( F_{context}(C_{k'}) \right)} \approx \alpha_k, \tag{11}$$

*where $F_{context}(C_k)$ is defined in Eq. 2. Therefore, the attention mechanism in Transformers forms a distribution that approximates $\boldsymbol{\pi} = P(\boldsymbol{\phi}, F_{context}(\boldsymbol{C})) = \{\pi_k\}_{k=1}^{K}$.*

**Theorem 3.** *In the case of $K > 1$, a mixed multinomial-based logic semantic distribution $P_{MMD}$ can be established by combining $P(\boldsymbol{\phi}, F_{context}(\boldsymbol{C}))$ and $P(\boldsymbol{\theta}, F_{rule}(\boldsymbol{z} \mid \boldsymbol{C}))$ as:*

$$
\begin{aligned}
P_{MMD} &= P(\boldsymbol{\phi}, F_{context}(\boldsymbol{C}), \boldsymbol{\theta}, F_{rule}(\boldsymbol{z} \mid \boldsymbol{C})) \\
&= P(\boldsymbol{\phi}, F_{context}(\boldsymbol{C})) \cdot P(\boldsymbol{\theta}, \boldsymbol{\phi}, F_{rule}(\boldsymbol{z} \mid \boldsymbol{C})) \\
&= \sum_{k=1}^{K} \pi_k \cdot P_{MD} = \sum_{k=1}^{K} \frac{1}{Z(\boldsymbol{\theta})} \frac{N!}{\prod_{j=1}^{J} n_j!} \exp\left( \log \pi_k + \sum_{j=1}^{J} \theta_j \cdot n_j \cdot F_{rule}(z_j \mid C_k) \right),
\end{aligned} \tag{12}
$$

*where $P(\boldsymbol{\theta}, \boldsymbol{\phi}, F_{rule}(\boldsymbol{z} \mid \boldsymbol{C}))$ follows the form provided in Eq. 8 in Theorem 1. The term $F_{rule}(z_j \mid C_k)$ is defined as in Eq. 4.*

Due to the page limit, the detailed proofs of all the theorems in the main text are delegated in the appendix.

**Remark 1.** ***Traditional MLN and MD-based NSL as Special Cases of MMD-NSL***: *Traditional MLN ($P_{MLN}$) are encompassed within our framework as a Logic Semantic Distribution characterized by weighted logical relationships. The MD-based NSL, a special case of $P_{MLN}$, occurs when $K = 1$. In both cases, the bias term $\log(\pi_k)$ does not affect the distribution, as there is no additional categorization of subpopulations, thus focusing solely on the logical structure without contextual differentiation.*

**Remark 2.** ***Mixed Multinomial Logic Semantic Distribution Weight Structure Analogous to Neural Networks***: *The structure of the weights $\theta_j$ and the bias terms $\log \pi_k$ in the Mixed Multinomial Logic Semantic Distribution closely resembles the weights and biases found in neural networks. This analogy highlights how the mixing coefficient and rule-specific weights can be viewed as learning parameters that influence the continuous relaxation of logical and contextual variables, similar to neural network layers.*

**Remark 3.** ***Unified Semantic through KG Embeddings***: *Both the fuzzy logic semantic and context logic semantic, as defined in Theorem 1, can be unified within the semantic embeddings mapping by $\{\mathbf{e}_{NER_{head}}, \mathbf{e}_{NER_{tail}}, \mathbf{e}_{path}\}$ of $\mathcal{G}$. This unification is achieved by leveraging embeddings to represent both rule-based logic and semantic context in a continuous space, enabling joint reasoning over logical structures and categorical contexts.*

## 3.3 OPTIMIZATION

The overall optimization objective is to maximize the log-likelihood of the Mixed Multinomial Logic Semantic Distribution $P_{MMD}$. This distribution combines both the context-dependent logic semantic

and the rule-based logic semantic into a unified probabilistic framework:

$$\log P_{\text{MMD}} = \log P(\boldsymbol{\phi}, F_{\text{context}}(\boldsymbol{C}), \boldsymbol{\theta}, F_{\text{rule}}(\boldsymbol{z} \mid \boldsymbol{C})). \tag{13}$$

In this work, we utilize a bilevel optimization framework due to the nested dependencies between the mixing coefficient $\boldsymbol{\pi}(\boldsymbol{\phi})$ and the logical rule weights $\boldsymbol{\theta}$. The bilevel structure allows us to jointly model both the context categories (which depend on $\boldsymbol{\phi}$) and the rule dependencies (captured by $\boldsymbol{\theta}$). These two aspects are interdependent, as the optimal rule weights depend on the contextual semantic and vice versa. This joint optimization is necessary to learn complex semantic categories from NER-type combinations and to weight logical rules effectively for capturing long-range dependencies within KG.

The bilevel optimization is expressed in Eq. 14:

$$\min_{\boldsymbol{\phi}} \; \ell(\boldsymbol{\phi}, \boldsymbol{\theta}^{\star}(\boldsymbol{\phi}); \boldsymbol{z} \mid \boldsymbol{C}) \tag{14a}$$

$$\text{subject to} \quad \theta_k^{\star}(\boldsymbol{\phi}) \in \arg\max_{\theta_k} \left( \frac{1}{Z(\boldsymbol{\theta})} \sum_{j=1}^{J} \theta_{jk} \cdot n_j \cdot F_{\text{rule}}(z_j \mid C_k) \right), \quad \forall k \in \{1, \ldots, K\}. \tag{14b}$$

In Eq. 14a, we maximize the overall objective $\ell(\boldsymbol{\phi}, \boldsymbol{\theta}^{\star}(\boldsymbol{\phi}); \boldsymbol{z} \mid \boldsymbol{C})$ by optimizing $\boldsymbol{\phi}$, which represents the parameters of the transformer model $T_{\boldsymbol{\phi}}(\boldsymbol{z} \mid \boldsymbol{C})$. The use of a transformer is appropriate here because the attention mechanism within the transformer naturally aligns with the computation of the mixing coefficient $\boldsymbol{\pi}(\boldsymbol{\phi})$, effectively determining the importance of different semantic contexts. The inputs to the transformer include the logical rule bodies $r_1^j \wedge \ldots \wedge r_L^j$ and the context triplet $\langle C_k(h), r_{\text{head}}, C_k(t) \rangle$, and the loss function, $\ell$, is designed to measure how well these rules are satisfied in the graph $\mathcal{G}$. This upper-level optimization updates $\boldsymbol{\phi}$, allowing the model to learn context representations via attention.

Simultaneously, the lower-level problem (Eq. 14b) focuses on maximizing the context-specific log-likelihood of the Multinomial Logic Semantic Distribution $P_{\text{MD}}$ with respect to $\theta_k$. The constant term $\frac{N!}{\prod_{j=1}^{J} n_j!}$ is excluded from the optimization process, as it does not affect the gradients or the optimization of $\theta_k$, which are the weights assigned to the rules within each context $C_k$. By maximizing the summation of the weighted rule scores, the model learns optimal weights $\theta_k^{\star}(\boldsymbol{\phi})$ for each context $k$, effectively capturing the logical dependencies in the KG that are influenced by the context semantic learned at the upper level.

### 3.4 Algorithm

The algorithm 1 adopts a bilevel optimization framework, where the lower and upper levels are optimized iteratively to capture both semantic categories and logical dependencies within KG.

At the lower level, for each context $C_k$, the current transformer parameters $\boldsymbol{\phi}$ (obtained from the upper level) are used to generate candidate rule bodies. Specifically, the transformer encodes the triplet $\langle C_k(h), r_{\text{head}}, C_k(t) \rangle$ into the embedding layer using unique indices. It then iteratively generates $r_1$ to $r_L$, step by step, to form a rule body $r_1 \wedge \ldots \wedge r_L$. This transformer inference process is performed multiple times to generate multiple candidate rule bodies. These candidate rule bodies may include duplicates, which are filtered using a simple uniqueness function. The function extracts $J$ unique rule bodies $r_1^j \wedge \cdots \wedge r_L^j$ corresponding to the query $r_{\text{head}}$, while also returning the count $n_j$ for each $j$-th unique rule body. The lower-level optimization focuses on maximizing the log-likelihood of the Multinomial Logic Semantic Distribution $P_{\text{MD}}$, as outlined in Eq. 14b. Specifically, the context-specific weights $\theta_k$ are optimized by maximizing the summation of the weighted rule scores. This optimization step allows the model to learn the optimal weights $\theta_k^{\star}(\boldsymbol{\phi})$ for each rule, effectively capturing the logical dependencies within each context $k$ in the KG $\mathcal{G}$.

In the upper-level optimization, after updating the rule weights $\theta_k^{\star}(\boldsymbol{\phi})$ in the lower level, the goal is to maximize the objective $\ell(\boldsymbol{\phi}, \boldsymbol{\theta}^{\star}(\boldsymbol{\phi}))$, as given in Eq. 14a. This objective is designed to update the transformer parameters $\boldsymbol{\phi}$, which encode the semantic embeddings and contextual representations. The transformer model $T_{\boldsymbol{\phi}}$ embeds the input rule bodies $r_1^j \wedge \ldots \wedge r_L^j$ along with their associated context triplet $\langle C_k(h), r_{\text{head}}, C_k(t) \rangle$. The attention mechanism within the transformer aligns with

the mixing coefficient $\pi_k(\phi)$, thereby determining the relative importance of different contexts. The loss function $\ell$ measures how well the generated rule bodies align with the constraints in $\mathcal{G}$, refining both the context embeddings and attention weights throughout the optimization process.

---

**Algorithm 1** Bilevel Optimization for `MMD-NSL`

---

1: **Initialize** parameters: transformer weights $\phi$, rule weights $\theta$
2: **for each iteration do**
3:     `// Lower-level optimization`
4:     **for each context** $C_k$ **do**
5:       Generate rule bodies $r_1^j \wedge \cdots \wedge r_L^j$ using current $\phi$.
6:       Solve Eq. 14b: maximize $\frac{1}{Z(\theta)} \sum_{j=1}^{J} \theta_{jk} \cdot n_j \cdot F_{\text{rule}}(z_j \mid C_k)$.
7:       Update $\theta_k^\star(\phi)$ for context $C_k$.
8:     **end for**
9:     `// Upper-level optimization`
10:     Solve Eq. 14a: maximize $\ell(\phi, \theta^\star(\phi))$ using transformer $T_\phi$.
11:     Update $\phi$ with the generated rule samples and their associated probabilities.
12: **end for**
13: **Return**: Optimized parameters $\phi$ and $\theta$.

---

## 4 EXPERIMENT

For evaluating our `MMD-NSL`, we use a KG relationship predictor as downstream task to assess algorithm 1. Each relation query to be predicted is treated as a rule query (rule head). Our NSL rule learner generates relevant rule bodies by sampling from the multiple NER-pair context within the NSL model, instead of relying on exhaustive path searches through a large KG. These newly discovered rule bodies provide a more accurate representation of the rule head (i.e., the relation query), leading to more precise label predictions for the relation query.

### 4.1 DATA PROBABILISTIC RECOMPILATION

Our objective is to evaluate the diversity of rules across different contexts. The dataset used is from DWIE (Document-Level Web Information Extraction) Zaporojets et al. (2021), comprising 799 documents categorized into 10 NER types and 65 relationship categories. For consistency checks, we utilized 39 golden first-order logic predicates from the DWIE dataset, including atomic formulas such as $player\_of(X,Y) \leftarrow member\_of(X,Y) \wedge sport\_player(X)$. The dataset was restructured into a dictionary format, where each key is a triplet $\langle$NER(head), rule_head, NER(tail)$\rangle$, and the corresponding value is a set of rule bodies paired with their frequencies, represented as rule probabilities. This compilation resulted in 23 sub-datasets identified by different rule heads, with each sub-dataset containing multiple NER combinations that share the same rule head.

### 4.2 MODEL OPTIMIZATION SETUP

At the upper level, the model employs a transformer-based architecture to learn a Context Logic Semantic Distribution. The input to the transformer includes the NER combination and the rule head, with the self-attention mechanism used to compute a continuous mixed coefficient representation while encoding the rule query (*rule_head*). This allows the model to represent categories and relations using unique numerical identifiers. The embedding layer for relations has dimensions of $(256, 2R + 1)$, where $R$ represents the total number of relations (65 in this case). Similarly, the NER category embedding layer has dimensions of $(256, 10)$, corresponding to 10 distinct NER categories. The model architecture includes two encoding and decoding layers, with an output layer of size $(256, 2R + 1)$. The input to the transformer is constructed by concatenating the rule head and rule body, each of size 4. If the input length is insufficient, padding symbols are added and masked using a $4 \times 4$ positional mask. On average, each rule head generates 50 candidate rule bodies. These candidates are filtered to remove duplicates and then passed to the lower-level model for further processing. At the lower level, the model dynamically initializes weights for each training round. These weights are specifically tailored to the candidate rule bodies generated by the upper level, capturing

Figure 2: A probabilistic heatmap illustrating a mixed multinomial logic semantic distribution for one sub-dataset out of 23. The heatmap represents the top 3 highest probabilities of different rule bodies for the rule query (*rule head: member_of*) evaluated across six distinct NER combination contexts.

their variability, which arises from the stochastic optimization process and the probabilistic nature of the generated rules. The size of each weight group is represented as $(23, J \times 1)$, where 23 denotes the number of rule head categories. Each member of the weight group corresponds to a candidate rule body and has dimensions of $(J \times 1)$. This hierarchical structure enables the model to effectively handle both rule-specific and contextual variations within the optimization process.

## 4.3 Downstream Task Evaluation

We selected four closely related and representative works that play a significant role in the theoretical foundation of our `MMD-NSL`. These works serve as baselines: LTN represents a fuzzy logic-based extension of MLNs, corresponding to Eq. 3. NMLN, a neural network-based MLN extension, shares a similar structure with Eq. 12 as noted in Remark 2. RNNLogic and LogiRE are special cases with multinomial distribution models as described by Eq. 8.

Our evaluation aims to determine whether `MMD-NSL` provides a more generalized framework by integrating the traditional MLN capability for handling long dependency chains with context modeling from the context semantic distribution to manage complex categorization effectively. This integration results in a more robust and adaptable paradigm. Therefore, we conducted a fine-grained performance analysis at the level of individual rule queries across different NER pair contexts. We compared `MMD-NSL` against these four baseline models, evaluating performance over 23 sub-datasets, each grouped by distinct rule heads. Each sub-dataset contains a unique rule head with multiple NER pair combinations, enabling an assessment of model performance across various relational contexts using the F1 score as the primary metric. As shown in Table 1, while certain MLN-based and multinomial-based methods marginally outperformed `MMD-NSL` on a few specific rule head sub-datasets, `MMD-NSL` consistently demonstrated superior performance across the majority of sub-datasets. This highlights its strength as a more comprehensive and adaptable approach, effectively managing both long dependency chains and complex categorization.

## 4.4 Rule Probabilistic Visualization

To provide a clear understanding of mixed multinomial logic semantic distributions, we present visualizations illustrating how the same rule head query can result in distinct rule_body outcomes depending on different NER pair combination contexts. The self-recursive rule, such as *member_of* ← *member_of*, exhibits the highest probability in most contexts, aligning with the characteristics of probability distributions. However, an exception occurs in the organization-organization NER context, where the rule head (*member_of*) reflects a relational pattern that represents membership facilitated through intermediary entities. This indicates that the *member_of* rule body reasoning is influenced by contextual semantics.

Beyond self-recursive rules, a comparison between the person-person and person-organization NER contexts highlights significant variations in rule bodies depending on the context. For instance, rules such as *head_of* and *citizen_of* ∧ *based_in* are meaningful for linking a person to an organization, whereas rules like *spokesperson_of* and *citizen_of* ∧ ∼*citizen_of* are relevant for linking a person to another person. These differences underline the importance of learning mixed multinomial logic semantic distributions that account for contextual variations, enabling more nuanced and accurate reasoning across diverse contexts.

Table 1: Downstream Task Performance Comparison

| rule_head | F1-score | | | | |
|---|---|---|---|---|---|
| | MLN-based representative | | Multinomial-based representative | | Mixed Multinomial |
| | LTN | NMLN | RNNLogic | LogiRE | `MMD-NSL` |
| citizen_of | 0.6614 | 0.7277 | 0.6751 | 0.6736 | **0.7470** |
| in0 | 0.7467 | 0.7645 | 0.7466 | 0.7569 | **0.7819** |
| in0-x | 0.6667 | 0.6783 | 0.6979 | **0.7090** | 0.6886 |
| gpe0 | 0.7742 | 0.8108 | 0.7586 | 0.7806 | **0.8268** |
| member_of | 0.1754 | 0.5759 | 0.5964 | **0.6226** | 0.6165 |
| agent_of | 0.5240 | 0.7090 | 0.6186 | 0.6257 | **0.7160** |
| citizen_of-x | 0.6448 | 0.6998 | 0.6667 | 0.6641 | **0.7034** |
| based_in0 | 0.6162 | 0.7281 | 0.6667 | 0.6711 | **0.7399** |
| based_in0-x | 0.6195 | 0.7458 | 0.6456 | 0.6441 | **0.7495** |
| head_of | 0.1340 | 0.3886 | 0.5422 | **0.6029** | 0.5672 |
| minister_of | 0.2314 | 0.6071 | 0.6250 | 0.6720 | **0.6769** |
| minister_of-x | 0.2311 | 0.5070 | 0.7907 | **0.8889** | 0.8818 |
| based_in2 | 0.0544 | 0.2432 | 0.2373 | 0.2222 | **0.3429** |
| head_of_state | 0.3423 | 0.5500 | **0.5789** | 0.5753 | **0.5789** |
| head_of_state-x | 0.2250 | 0.6098 | 0.5859 | 0.5979 | **0.6494** |
| agency_of | 0.2827 | 0.4793 | 0.5051 | 0.5679 | **0.6207** |
| agency_of-x | 0.2129 | 0.5487 | 0.6087 | 0.6067 | **0.6667** |
| in2 | 0.0717 | 0.4286 | 0.5556 | **0.5625** | 0.5200 |
| event_in0 | 0.3429 | **0.4138** | 0.3636 | 0.3636 | 0.3571 |
| award_received | **0.5533** | 0.5154 | 0.5263 | 0.4706 | 0.5455 |
| appears_in | 0.4242 | 0.4819 | **0.5181** | 0.5000 | 0.4872 |
| vs | 0.2524 | 0.2927 | 0.1523 | 0.1659 | **0.3057** |
| won_vs | 0.0436 | 0.0902 | 0.0456 | 0.0535 | **0.0976** |
| spokesperson_of | 0.0016 | 0.0392 | 0.0000 | 0.0000 | **0.0909** |
| created_by | 0.0061 | 0.2222 | 0.3000 | **0.3167** | 0.2222 |
| event_in2 | 0.0000 | 0.0000 | 0.1250 | **0.1857** | 0.1500 |

## 5  CONCLUSIONS

In this paper, we introduced `MMD-NSL`, a novel framework for NSL that unifies the handling of long dependency chains and complex semantic categorization within KG. By leveraging a continuous Mixed Multinomial Logic Semantic Distribution, we extended traditional MLN to incorporate context-dependent semantic embeddings. Our theoretical contributions, including the bijective mapping between MLNs and continuous multinomial distributions, establish a foundation for capturing intricate dependencies and diverse contexts in NSL tasks. The framework employs a bilevel optimization process, where the transformer-based upper level efficiently learns mixing coefficient analogous to attention mechanisms, and the lower level optimizes rule weights, allowing for effective learning of both context and rule patterns. Experimental results show that `MMD-NSL` provides a more general and adaptable paradigm for NSL, outperforming traditional baselines in managing complex relationships and multiple semantic contexts, thereby advancing continuous probabilistic models for neuro-symbolic reasoning and complex relational tasks.

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

## A  APPENDIX

In this material, we provide more detailed discussions on the theory and realization of our `MMD-NSL`.

### A.1  PROOF OF THEOREM 1

*Proof.* In the case of $K = 1$, we aim to establish a bijective mapping from the MLN-based logic semantic distribution in Eq. 1 to the multinomial distribution-based logic semantic distribution in Eq. 8. We begin by recalling the MLN-based logic semantic distribution:

$$P_{\text{MLN}}(\boldsymbol{z}) = \frac{1}{Z} \exp\left(\sum_{j=1}^{J} w_j f_j(z_j)\right) \quad \text{(Eq. 1)}, \tag{15}$$

where $Z$ is the partition function, $w_j$ are weights, and $f_j(z_j)$ are feature functions.

**Step 1:** Define the fuzzy logic semantic function $F_{\text{rule}}(z_j)$ as per Eq. 3:

$$F_{\text{rule}}(z_j) = \frac{1}{2} y(r_{\text{head}}) \cdot \max_{\mathcal{G}} \prod_{l=1}^{L} \mathbf{e}_{\text{path}}(r_l) \quad \text{(Eq. 3)}. \tag{16}$$

This function maps the discrete logical variable $z_j$ into a continuous value in $\mathbb{R}$.

**Step 2:** Relate the counts $n_j$ to the variables $z_j$. Since $z_j = 1$ if the rule $z_j$ holds true in $\mathcal{G}$ and $-1$ otherwise, $n_j$ represents the count of true groundings of rule $z_j$ in $N$ samples.

**Step 3:** Define the multinomial logic semantic distribution $P_{\text{MD}}(\boldsymbol{z})$ as:

$$P_{\text{MD}}(\boldsymbol{z}) = \frac{1}{Z(\boldsymbol{\theta})} \frac{N!}{\prod_{j=1}^{J} n_j!} \exp\left(\sum_{j=1}^{J} \theta_j \cdot n_j \cdot F_{\text{rule}}(z_j)\right) \quad \text{(Eq. 8)}, \tag{17}$$

where $\boldsymbol{\theta} = \{\theta_j\}_{j=1}^{J}$ are the weights associated with each rule $z_j$, and the partition function $Z(\boldsymbol{\theta})$ is given by:

$$Z(\boldsymbol{\theta}) = \left(\sum_{j=1}^{J} e^{\theta_j \cdot F_{\text{rule}}(z_j)}\right)^N. \tag{18}$$

**Step 4:** Establish the mapping between the MLN and multinomial distributions by aligning the parameters and functions:

$$w_j = \theta_j, \quad f_j(z_j) = n_j \cdot F_{\text{rule}}(z_j), \quad Z = Z(\boldsymbol{\theta}) \frac{\prod_{j=1}^{J} n_j!}{N!}. \tag{19}$$

**Step 5:** Consider the multinomial distribution over $J$ categories with counts $\{n_j\}$ and probabilities $\{p_j\}$:

$$P(\{n_j\}; \{p_j\}) = \frac{N!}{\prod_{j=1}^{J} n_j!} \prod_{j=1}^{J} p_j^{n_j}. \tag{20}$$

**Step 6:** Parameterize the probabilities $p_j$ using $\theta_j$ and $F_{\text{rule}}(z_j)$:

$$p_j = \frac{e^{\theta_j \cdot F_{\text{rule}}(z_j)}}{\sum_{k=1}^{J} e^{\theta_k \cdot F_{\text{rule}}(z_k)}}. \tag{21}$$

**Step 7:** Substitute $p_j$ back into the multinomial distribution:

$$P(\{n_j\}; \boldsymbol{\theta}) = \frac{N!}{\prod_{j=1}^{J} n_j!} \left(\frac{1}{Z(\boldsymbol{\theta})} \prod_{j=1}^{J} e^{\theta_j \cdot n_j \cdot F_{\text{rule}}(z_j)}\right), \tag{22}$$

where the partition function $Z(\boldsymbol{\theta})$ is:

$$Z(\boldsymbol{\theta}) = \left( \sum_{k=1}^{J} e^{\theta_k \cdot F_{\text{rule}}(z_k)} \right)^{N}. \tag{23}$$

**Step 8:** Simplify the expression to match the form of $P_{\text{MD}}(\boldsymbol{z})$:

$$P(\{n_j\}; \boldsymbol{\theta}) = \frac{1}{Z(\boldsymbol{\theta})} \frac{N!}{\prod_{j=1}^{J} n_j!} \exp\left( \sum_{j=1}^{J} \theta_j \cdot n_j \cdot F_{\text{rule}}(z_j) \right). \tag{24}$$

**Step 9:** Compare with the MLN-based distribution and confirm the bijective mapping by recognizing that the exponents and normalization factors align when parameters are identified as per Step 4.

**Conclusion:** We have established that the multinomial logic semantic distribution $P_{\text{MD}}(\boldsymbol{z})$ in Eq. 8 can be mapped bijectively to the MLN-based logic semantic distribution $P_{\text{MLN}}(\boldsymbol{z})$ in Eq. 1 by appropriately defining the weights $w_j = \theta_j$, feature functions $f_j(z_j) = n_j \cdot F_{\text{rule}}(z_j)$, and the partition function $Z$. This completes the proof of Theorem 1. $\square$

### A.2 PROOF OF THEOREM 2

*Proof.* **Step 1: Define the Attention Mechanism**

The attention weights $\alpha_k$ in a Transformer are given by the scaled dot-product attention:

$$\alpha_k = \frac{\exp\left( \frac{Q^\top K_k}{\sqrt{d}} \right)}{\sum_{k'=1}^{K} \exp\left( \frac{Q^\top K_{k'}}{\sqrt{d}} \right)}, \tag{25}$$

where: - $Q \in \mathbb{R}^d$ is the query vector. - $K_k \in \mathbb{R}^d$ is the key vector associated with the $k$-th context. - $d$ is the dimensionality of the vectors.

**Step 2: Define Query and Key Vectors in Terms of NER and Context Embeddings**

Let:

$$Q = \mathbf{e}_{\text{NER}_{\text{head}}} \odot \mathbf{e}_{\text{NER}_{\text{tail}}}, \tag{26}$$
$$K_k = \mathbf{e}_{C_k}, \tag{27}$$

where $\mathbf{e}_{\text{NER}_{\text{head}}} \odot \mathbf{e}_{\text{NER}_{\text{tail}}}$ denotes the element-wise product of the head and tail NER embeddings, and $\mathbf{e}_{C_k}$ is the context-specific embedding vector.

**Step 3: Recall the Context Logic Semantic Function**

The context logic semantic function $F_{\text{context}}(C_k)$ is defined as:

$$F_{\text{context}}(C_k) = \mathbf{e}_{C_k}^\top \left( \mathbf{e}_{\text{NER}_{\text{head}}} \odot \mathbf{e}_{\text{NER}_{\text{tail}}} \right). \tag{28}$$

**Step 4: Compute the Attention Scores**

The unnormalized attention scores $u_k$ between the query vector $Q$ and the key vector $K_k$ are:

$$u_k = \frac{Q^\top K_k}{\sqrt{d}} = \frac{\left( \mathbf{e}_{\text{NER}_{\text{head}}} \odot \mathbf{e}_{\text{NER}_{\text{tail}}} \right)^\top \mathbf{e}_{C_k}}{\sqrt{d}}. \tag{29}$$

By the definition of $F_{\text{context}}(C_k)$, we have:

$$u_k = \frac{F_{\text{context}}(C_k)}{\sqrt{d}}. \tag{30}$$

**Step 5: Compute the Attention Weights**

The attention weights $\alpha_k$ are computed as:

$$\alpha_k = \frac{\exp\left(\frac{F_{\text{context}}(C_k)}{\sqrt{d}}\right)}{\sum_{k'=1}^{K} \exp\left(\frac{F_{\text{context}}(C_{k'})}{\sqrt{d}}\right)}. \tag{31}$$

**Step 6: Relate Attention Weights to Context Logic Semantic Distribution**

The context logic semantic distribution $\pi$ is defined as:

$$\pi_k = \frac{\exp\left(F_{\text{context}}(C_k)\right)}{\sum_{k'=1}^{K} \exp\left(F_{\text{context}}(C_{k'})\right)}. \tag{32}$$

**Step 7: Align the Scaling Factor**

To align $\alpha_k$ with $\pi_k$, we adjust for the scaling factor $\sqrt{d}$ in the attention mechanism. Define a scaled context function:

$$\tilde{F}_{\text{context}}(C_k) = \frac{F_{\text{context}}(C_k)}{\sqrt{d}}. \tag{33}$$

Then the attention weights become:

$$\alpha_k = \frac{\exp\left(\tilde{F}_{\text{context}}(C_k)\right)}{\sum_{k'=1}^{K} \exp\left(\tilde{F}_{\text{context}}(C_{k'})\right)}, \tag{34}$$

which approximates $\pi_k$ up to the scaling factor.

**Step 8: Conclusion**

By redefining the query and key vectors using NER embeddings and context-specific embeddings, and by appropriately scaling the context function, the attention weights computed by the Transformer are equivalent to the Context Logic Semantic Distribution $\pi$. Therefore, the attention mechanism forms a Context Logic Semantic Distribution as defined in Eq. 32.

$\square$

A.3 PROOF OF THEOREM 3

*Proof.* **Step 1: Consider the Mixed Multinomial Logic Semantic Distribution**

From Eq. 12, the mixed multinomial logic semantic distribution is defined as:

$$P_{\text{MMD}} = \sum_{k=1}^{K} \frac{1}{Z(\boldsymbol{\theta})} \frac{N!}{\prod_{j=1}^{J} n_j!} \exp\left(\log \pi_k + \sum_{j=1}^{J} \theta_j \cdot n_j \cdot F_{\text{rule}}(z_j \mid C_k)\right). \tag{35}$$

Here, $\pi_k$ represents the softened mixing coefficients defined in Eq. 11, and $F_{\text{rule}}(z_j \mid C_k)$ is given by Eq. 4.

**Step 2: Combine Context and Rule Contributions**

The term $\log \pi_k$ incorporates the context information via $F_{\text{context}}(C_k)$, as per Eq. 11:

$$\pi_k = \frac{\exp\left(F_{\text{context}}(C_k)\right)}{\sum_{k'=1}^{K} \exp\left(F_{\text{context}}(C_{k'})\right)}. \tag{36}$$

This reflects the influence of different contexts $C_k$ on the overall distribution.

**Step 3: Align Feature Functions and Weights**

As in the proof of Theorem 1, we set:

$$f_j(z_j \mid C_k) = n_j \cdot F_{\text{rule}}(z_j \mid C_k), \quad w_j = \theta_j. \tag{37}$$

This aligns the rule-specific contributions in both the MLN and mixed multinomial distributions.

**Step 4: Express the MLN Distribution with Contexts**

The MLN-based logic semantic distribution incorporating contexts becomes:

$$P_{\text{MLN}}(z) = \frac{1}{Z} \exp \left( \sum_{k=1}^{K} \sum_{j=1}^{J} w_j f_j(z_j \mid C_k) + \sum_{k=1}^{K} \phi_k F_{\text{context}}(C_k) \right), \tag{38}$$

where $\phi_k$ are weights associated with each context $C_k$.

**Step 5: Recognize the Mixture Structure**

The mixed multinomial distribution $P_{\text{MMD}}$ effectively represents a mixture model over contexts:

$$P_{\text{MMD}} = \sum_{k=1}^{K} \pi_k P_k(z), \tag{39}$$

where each $P_k(z)$ is a multinomial distribution conditioned on context $C_k$.

**Step 6: Conclude the Establishment of $P_{\text{MMD}}$**

By combining the context contributions $P(\phi, F_{\text{context}}(C))$ with the rule contributions $P(\theta, F_{\text{rule}}(z \mid C))$, and aligning the feature functions and weights, we confirm that $P_{\text{MMD}}$ can be expressed as in Eq. 12. This demonstrates that a mixed multinomial-based logic semantic distribution is established by combining the context and rule-based distributions, as stated in the theorem.

$\square$

