# OpenReview forum: "MMD-NSL: Mixed Multinomial Distribution-based Neuro-Symbolic Learning"
_ICLR.cc/2025/Conference — Submitted to ICLR 2025_

### Official Review · Reviewer_HreQ · 2024-11-02

**Soundness:** 2
**Presentation:** 1
**Contribution:** 2
**Rating:** 5
**Confidence:** 3

**Summary:**

This paper introduces the MMD-NSL framework, which extends classic MLNs by incorporating context-aware semantic embeddings, in order to both handling long dependency chains and addressing the complexity of semantic categorization within KGs. This is achieved by introducing a continuous Mixed Multinomial Logic Semantic Distribution. The framework exploit a bi-level optimization exploiting transformer-based attention mechanism in an upper level, while also learning rule weights at the lower level. MMD-NSL shows significant improvements wrt existing sota approaches across several datasets.

**Strengths:**

The aim and the idea behind the paper seem interesting and the experimental results are good. However I found the paper too confusing, and it is very difficult to assess the concrete contribution of this paper in its current form (see weaknesses and questions).

**Weaknesses:**

- Related work discussion is quite limited, for a topic that has been extensively studied by different spots. For instance, modelling logic rules as latent variables has been already considered e.g. in [1] or [2], and also extending MLNs with neural components [3]. Similarly for what concern logic rules in the context of KGs, sota approaches like RLogic [4] or NCRL [5] should be discussed, or also LERP [6] that models contextual information from neighboring sub-graphs of entities in logic rules.
- Too many symbols and statements come out from nowhere, without having been introduced or explained. See both other comments and questions for more details.

OTHER COMMENTS
- "Typically, NSL tasks consider long dependencies expressed as simple logical implications like r_head → r_1 ∧ . . . ∧ r_L" generally is the opposite, sevearl NSL systems rely on Horn clauses of the form r_1 ∧ . . . ∧ r_L → r_head. The implication here has a different meaning, or what the authors mean?
- "We generalize this to tasks formulated as ⟨C(h), rhead,C(t)⟩ → r1 ∧ . . . ∧ rL, where C(·) denotes the NER types of the node", what are "h" and "t"? What is the NER type of  node? These symbols and notions have not been defined before.
- "the first-order logic rule bodies r1 ∧ . . . ∧ rL are", generally the body is a conjunction of literals (e.g. atoms in FOL), hence here I saw only ONE body with L relations. Is there a typo or what the authors mean by body exactly? These terms, as well as what the r_i mean have never been explained.
- Definition 1 is unclear. So you call a Logic Semantic Variable any continuous image by a "relaxation function" F to  a vairable zj based on the embeddings {eNERhead , eNERtail , epath} of G? What are these embeddings?, How they are obtained? What is meant by "based on"? This statement cannot be a formal definition without really stating/formally declaring the mentioned objects that are referred.
- "an semantic" typo
- "LTN represents a fuzzy logic-based extension of MLNs" LTN is not a fuzzy extension of MLN. In LTN the weights of the rules are fixed and cannot be learnt. A fuzzy extension of MLN is given e.g. by PSL [7].

References
[1] Maene, Jaron, and Luc De Raedt. "Soft-unification in deep probabilistic logic." Advances in Neural Information Processing Systems 36 (2024).
[2] Marra, Giuseppe, Michelangelo Diligenti, and Francesco Giannini. "Relational reasoning networks." arXiv preprint arXiv:2106.00393 (2021).
[3] Marra, Giuseppe, et al. "Relational neural machines." ECAI 2020. IOS Press, 2020. 1340-1347.
[4] Cheng, Kewei, et al. "Rlogic: Recursive logical rule learning from knowledge graphs." Proceedings of the 28th ACM SIGKDD Conference on Knowledge Discovery and Data Mining. 2022.
[5] Cheng, Keiwei, Nesreen K. Amed, and Yizhou Sun. "Neural Compositional Rule Learning for Knowledge Graph Reasoning." International Conference on Learning Representations (ICLR). 2023.
[6] Han, Chi, et al. "Logical Entity Representation in Knowledge-Graphs for Differentiable Rule Learning." The Eleventh International Conference on Learning Representations.
[7] Bach, Stephen H., et al. "Hinge-loss markov random fields and probabilistic soft logic." Journal of Machine Learning Research 18.109 (2017): 1-67.

**Questions:**

1) Can you make a clear example (and maybe it could be useful to add it to the main paper) about long dependency chains and complex
categorization that you mention all across the paper?
2) "We denote zj | Ck as the j-th rule variable conditioned on the k-th context variable, which specifically corresponds to the structured relation ⟨C(h), rhead,C(t)⟩ → r1 ∧. . .∧rL." Why C is not C_k and r_head r1, ..., rL do not depend on j?
3) "where zj = 1 if the rule zj holds true in G and −1 otherwise" what does it mean that the rule zj holds true in here? It is not explained if each rule is universally quantified or it is ground, wrt which variables or objects, what are the known facts, if it is assumed CWA.
4) What do denote f_j and z_j in equation 1? According to the beginning of section 3.2 z_j denotes a latent variable for a rule. How this conciliates with MLN where you have rules and each f_j(z_j) is just n_j number of groundings satisfying the rule. Or similarly in PSL with fuzzy relaxations or in NLMN with neural potentials?
5) "Following Definition 1, there are two types of Logic Semantic Variable:" Why only these two types?

---

> ### Author Response · Authors · 2024-11-25
>
> Thank you for your helpful comments and questions.
>
>
>
> **Response to Weakness 1:**
>
> We have added and discussed references [1]-[6] to the related works section, in section of related work  in red.
>
>  **Response to Comment 1:**
> We mean Horn clauses form, and we have revised it in Section 3.1, first paragraph, with a red mark.
>
>
> **Response to Comment 2:**
> We have clarified it in Section 3.1, first paragraph, with a red mark.
>
>
> **Response to Comment 3:**
>
> We have revised it in Section 3.1, first paragraph, with a red mark.
>
>
> **Response to Comment 4:**
> We have revised Definition 1 for clarity in revision version. All embeddings, including {eNERhead , eNERtail , epath}, are explicitly defined and used within these functions. Their roles and how they are obtained are explained in detail as part of the definitions of the respective functions.
>
>
> **Response to Comment 5:**
> We have corrected this typo in Remark 3, marked in red.
>
> **Response to Comment 6:**
>
> The categorization may vary depending on different interpretations. For instance, in the highly cited survey From Statistical Relational to Neuro-Symbolic Artificial Intelligence, LTN is categorized as extending MLNs, alongside Neural Markov Logic Networks (NMLN) and Relational Neural Machines (RNM). Specifically, the survey states:
>
> "Another group of approaches, including Logic Tensor Networks (LTN) [Donadello et al., 2017], Neural Markov Logic Networks (NMLN) [Marra and Kuželka, 2019], and Relational Neural Machines (RNM) [Marra et al., 2020], extend MLNs, allowing either predicates (LTN) or factors (NMLN and RNM) to be implemented as neural architectures."
>
> In contrast, PSL [7], which you mentioned, is categorized in the survey under a parallel class of Markov Logic Networks (MLNs), as the survey states: "The second category includes Markov Logic Networks (MLNs) [Richardson and Domingos, 2006] and Probabilistic Soft Logic (PSL) [Bach et al., 2017]. They essentially specify a set of weighted constraints, clauses or formulae."
>
>
> **Response to Question 1:**
>
> We have added a figure (Figure 1) in the main paper, highlighted in red, to illustrate an overview of the proposed framework. This figure explains the handling of long dependency chains and complex categorization through a detailed example, as described in the red-marked text within the introduction section.
>
> **Response to Question 2:**
> The formulas have been updated with changes marked in red.
>
>
> **Response to Question 3:**
> The known facts refer to rule samples (⟨C(h), r_head, C(t)⟩ → r1 ∧ . . . ∧ rL) in the KG training dataset. Whether a rule sample holds true indicates the ground truth label for the relation r_head under the context C(h)-C(t), specifying whether it is true (positive) or false (negative) based on the given rule body. All relations in the rule are quantified by relation embeddings, while NER types are quantified by NER embeddings. These rule samples are utilized in Eq.4, with further explanation text provided below Eq.4.
>
> **Response to Question 4:**
>
> Equation 1 can unify traditional MLNs with their extended variants. Specifically:
> 1. **Pure MLNs**: $f_j$ corresponds to $n_j$, the number of rule groundings, softened by a weight $w_j$.
> 2. **Fuzzy-Based MLN Extensions**: $f_j$ represents fuzzy operators, softened by a fuzzy score weight $w_j$.
>
>
> The inputs to $f_j$, which can include rule chains or logic rule formulas, are collectively denoted by $z_j$.
>
>
> **Response to Question 5:**
>
> Context Logic Semantic Variable in Eq.2 is inspired by the Dot-Product Attention mechanism. Consequently, in Theorem 2, the attention weights approximate the Context Logic Semantic Distribution composed of the Context Logic Semantic Variable.
>
> Fuzzy Logic Semantic Variable is inspired by fuzzy conjunction and fuzzy disjunction operations. These variables serve as a key link for establishing a bijective mapping between the MLN-based logic semantic distribution and the multinomial distribution-based logic semantic distribution in Theorem 1.
>
> Therefore, these two types of variables serve our paper’s aim of addressing context categorization and long dependency modeling. They form a necessary foundation for constructing a logic continuous distribution based on a mixed multinomial distribution, as discussed in Theorem 3.
>
> Our ultimate goal is to construct a unified logic continuous distribution based on a mixed multinomial distribution. This unified framework is achieved through the combination of the two types of Logic Semantic Variables.
>
> Other forms of Logic Semantic Variables could also be considered, provided they can be proven to form a mixed multinomial logic distribution.
>
> **We respectfully request reconsideration of our score.**

---

> > ### Comment · Reviewer_HreQ · 2024-11-27
> >
> > Dear authors,
> > thanks for you answer and clarifications, and I'm happy that you found my comments useful and that help you correcting typos and revising some statements. However, I still have some concerns on some of your answers. As a side note, I found a bit inelegant to ask explicitly for a reconsideration of your score. Any reviewer is perfectly knowing that the score could be increased.
> >
> > **LTN as an extension of MLN**:
> > I know very well the survey you're talking about, which distinguishes different dimensions to categorize the NeSy systems. For this reason the same systems appear different times within the text, and are grouped according to the different dimensions.
> > Moreover, the statement:
> > *"Another group of approaches, including Logic Tensor Networks (LTN) [Donadello et al., 2017], Neural Markov Logic Networks (NMLN) [Marra and Kuželka, 2019], and Relational Neural Machines (RNM) [Marra et al., 2020], extend MLNs, allowing either predicates (LTN) or factors (NMLN and RNM) to be implemented as neural architectures."*
> > refers to the distinction between directed and undirected graphical model, hence the term "extend" is (approximately) fine as it is related to this precise dimension.
> > Wrt this dimension, LTN and the others are in the same class of MLN, and LTN allows neural architectures (extend? it's a variant?). But, LTN does not allow learning the weight of the rules, LTN has a different semantics (fuzzy vs boolean), moreover LTN only considers MAP inference, and it has not a clear probabilistic semantics, etc.. Long story short, LTN does not allow to recover MLN as a special case (while this is done by RNM), thus if not limited to a specific aspect, it is incorrect to state that it is an extension.
> > For future reference, I also recommend you the new version of the survey you were referring to:
> > Marra, G., Dumančić, S., Manhaeve, R., & De Raedt, L. (2024). From statistical relational to neurosymbolic artificial intelligence: A survey. Artificial Intelligence, 104062.
> >
> > **Related work**:
> > I think that the related work section has improved after your integration. However, discussing the related work is not just listing some related systems, but it is mostly going a bit further in discussing the core differences of your approach wrt these systems. This should include also a bit more of experimental comparisons wrt these systems. I see your point about having "theorem-driven ablation experiments", but I also share some concerns of the other reviewers concerning the weakness of the comparison in the experiments.

---

### Official Review · Reviewer_wDkE · 2024-11-03

**Soundness:** 2
**Presentation:** 1
**Contribution:** 1
**Rating:** 3
**Confidence:** 4

**Summary:**

This work presents Neuro-Symbolic Learning based on Mixed Multinomial Distributions (MMD-NSL) with the aim of improving performance in handling complex relationships within Knowledge Graphs (KG). To this end, theoretical relations between the presented method and state-of-the-art methods are presented. The performance of MMD-NSL is evaluated on a synthetic and a real-world data benchmark.

**Strengths:**

1. The experiments show consistent improvements over state-of-the-art NeSy systems on a real-world dataset.

**Weaknesses:**

1. The presentation overall lacks clarity and needs further refinement. This includes many grammar errors and issues such as exact repetitions within the same page (e.g., "establishing a bijective mapping between MLNs and continuous multinomial distributions," both on lines 66 and 76).

2. The Figures do not support the significance of the contribution well. Figure 1 takes up half a page to demonstrate that a simple synthetic target could be learned. Figure 2 takes up an entire page with a questionable amount of information for the reader. Also, in Figure 2, labels are badly formatted (overlapping numbers) with misleading and inhomogeneous color mapping (e.g., a difference of 0.01 changing the hue completely). Again, some phrases are nearly identical, e.g., between Figure 1 caption and lines 442-446.

3. Given the nature of this contribution, namely a novel NeSy learning architecture, it would be highly desirable to compare performance across multiple datasets instead of a single one.

4. The mathematical presentation is, at times, lacking. For example, in the introduction and overview sections, logical rules are presented as a head implying the body (r_head -> r_1, r_2, ... ,r_n). Traditionally, the opposite is the case, with r_head <- r_1, r_2, ... ,r_n being the way Horn Clauses are written in implication form.

**Questions:**

Please see the weaknesses section

---

> ### Author Response · Authors · 2024-11-25
>
> Thank you for your helpful comments and questions.
>
>
> **Weakness 1:**
> *The presentation overall lacks clarity and needs further refinement. This includes many grammar errors and issues such as exact repetitions within the same page (e.g., "establishing a bijective mapping between MLNs and continuous multinomial distributions," both on lines 66 and 76).*
>
> **Response to Weakness 1:**
>
> In revision version, we have revised significant portions of the introduction to eliminate redundant sentences. Additionally, we have added a new figure (Figure 1) to visually explain the overall framework, improving clarity and addressing the need for further refinement. These changes are highlighted in red within the revised introduction section.
>
>
> **Weakness 2:**
> *The Figures do not support the significance of the contribution well. Figure 1 takes up half a page to demonstrate that a simple synthetic target could be learned. Figure 2 takes up an entire page with a questionable amount of information for the reader. Also, in Figure 2, labels are badly formatted (overlapping numbers) with misleading and inhomogeneous color mapping (e.g., a difference of 0.01 changing the hue completely). Again, some phrases are nearly identical, e.g., between Figure 1 caption and lines 442-446.*
>
> **Response to Weakness 2:**
>
> In revision version, we have removed the Synthetic Experiment section to allocate more space for detailed explanations of real-data experiments. Regarding the visualization in Figure 2, we have re-drawn the figure to address the issues you highlighted, including overlapping labels, misleading color mapping, and inhomogeneous formatting. Additionally, we have completely revised the section on rule probabilistic visualization to provide a more thorough explanation of the learning rules and their significance, ensuring that the updated figure effectively supports our contributions.
>
>
>
> **Weakness 3:**
> *Given the nature of this contribution, namely a novel NeSy learning architecture, it would be highly desirable to compare performance across multiple datasets instead of a single one.*
>
> **Response to Weakness 3:**
> We would like to emphasize two key points:
> 1. Our contribution is not solely focused on performance comparison with a single dataset. Instead, we evaluate using 23 NER-pair-aware datasets. Those fine gained dataset provides a rigorous evaluation approach.
> 2. Our research aims to better model complex NER-pair categorization rather than traditional long dependency considering baselines. Most used datasets do not offer sufficient meta-information to refine the data for rule-based NER-pair research, which is crucial for modeling complex NER-pair categorization effectively.
>
>
> **Weakness 4:**
> *The mathematical presentation is, at times, lacking. For example, in the introduction and overview sections, logical rules are presented as a head implying the body (r_head -> r_1, r_2, ..., r_n). Traditionally, the opposite is the case, with r_head <- r_1, r_2, ..., r_n being the way Horn Clauses are written in implication form.*
>
> **Response to Weakness 4:**
>
> It's a typo of arraw symbol, and we have revised it in revision version, : 'NSL tasks typically consider long dependencies expressed in the form $r_{\text{head}} \leftarrow r_1^j \wedge \ldots \wedge r_L^j$ , where $j $ represents the $ j $-th unique rule body for the current $r_{\text{head}} $, and $ L $ is the length of the relation along a rule body path.' in Section 3.1, first paragraph, with a red mark.
>
> **We respectfully request reconsideration of our score.**

---

> > ### Comment · Reviewer_wDkE · 2024-11-27
> > **Response**
> >
> > I would like to thank the authors for their response. I believe that the revised version provides more readability. I still believe that comparison to neuro-symbolic systmes is required as also pointed out by other reviewers. I would like to keep the score unchanged as of now.

---

### Official Review · Reviewer_RVFh · 2024-11-03

**Soundness:** 3
**Presentation:** 2
**Contribution:** 2
**Rating:** 3
**Confidence:** 4

**Summary:**

This paper develops an embedding-based, optimizable probabilistic framework that can be bijectively mapped to the MLN framework. Within this framework, the rule discovery and reasoning tasks are extended by incorporating the ontology information of entities in the rule head. A bilevel optimization strategy is employed to optimize the proposed model. Experimental results demonstrate the superiority of the proposed method.

**Strengths:**

1.	The bijective mapping between MLNs and continuous multinomial distributions is a valuable contribution.
2.	Incorporating ontology information into rule learning is a sensible approach.

**Weaknesses:**

1.	Ontology information is not fully utilized in the framework; it should also be applied to the tail part of the rule. This enhancement would make the paper more comprehensive and increase its contribution.
2.	The experimental design lacks a key analysis: how the two components—the ontology information and the MLN-based probabilistic framework—individually contribute to the observed improvements.
3.	Certain aspects of the algorithm are unclear. For instance: 1) How are the rules generated? 2) How is  n_j  calculated for each newly generated rule?

The paper addresses some important points in this field but does not yet meet the standards for the conference.

**Questions:**

Refer to the weaknesses section for the unclear points.

---

> ### Author Response · Authors · 2024-11-25
>
> Thank you for your helpful comments and questions.
>
> **Weakness 1:**
> Ontology information is not fully utilized in the framework; it should also be applied to the tail part of the rule. This enhancement would make the paper more comprehensive and increase its contribution.
>
> **Response to Weakness 1:**
> Our framework inherently addresses this concern through **Theorem 2**, which demonstrates that attention mechanisms can simultaneously encode ontology information for both the rule head and the rule body using a transformer architecture. Specifically:
> - **Self-attention** encodes ontology information related to the rule head.
> - **Encoder-decoder attention** further integrates the ontology information of the rule head while also encoding the rule body.
>
> This dual capability eliminates the need for explicit extensions to the rule tail, as the framework already captures and integrates ontology-related information effectively. By leveraging these mature and well-established mechanisms, our approach ensures comprehensive utilization of ontology information without additional modifications to the framework.
>
> **Weakness 2:**
> The experimental design lacks a key analysis: how the two components—the ontology information and the MLN-based probabilistic framework—individually contribute to the observed improvements.
>
> **Response to Weakness 2:**
> In Table 1, we have implemented and compared two baseline methods: one based purely on multinomial distributions without incorporating ontology information, and another using a purely MLN-based approach. These comparisons clearly demonstrate the individual contributions of each component, highlighting the observable improvements achieved by integrating ontology information and the MLN-based probabilistic framework within our method.
>
> **Weakness 3:**
> Certain aspects of the algorithm are unclear. For instance: 1) How are the rules generated? 2) How is n_j calculated for each newly generated rule?
> The paper addresses some important points in this field but does not yet meet the standards for the conference.
>
>
> **Response to Weakness 3:**
>
> In revision version, we have added a description of rule generation and the calculation of $ n_j $ in Section 3.4 (Algorithm), highlighted in red.
>
> **We respectfully request reconsideration of our score.**

---

### Official Review · Reviewer_b8oE · 2024-11-04

**Soundness:** 3
**Presentation:** 3
**Contribution:** 3
**Rating:** 6
**Confidence:** 4

**Summary:**

This paper presents the Mixed Multinomial Distribution-based NSL (MMD-NSL) framework, which adeptly addresses the challenge of complex categorization by seamlessly integrating the handling of long dependency chains with complex semantic categorization within KGs. Extensive experimental results demonstrate the effectiveness of the proposed method'.

**Strengths:**

1. The introduction of the Mixed Multinomial Distribution in the field of NSL for addressing complex classification issues exhibits novelty.
2. Both theoretical and empirical analyses are comprehensive, with theoretical findings establishing MMD-NSL as a more general form of classical NSL, thereby making a contribution to the NSL field.

**Weaknesses:**

In general, the experiments of the paper are relatively weak.  Firstly, there is a limited comparison of the proposed algorithm with other notable approaches such as DeepProblog [1] and Semantic Loss [2].  Secondly, despite the authors conducting experiments on numerous datasets, these datasets appear to be predominantly toy and simplistic.

References:
[1] DeepProbLog: Neural Probabilistic Logic Programming
[2] A Semantic Loss Function for Deep Learning with Symbolic Knowledge

**Questions:**

1. The author uses a bilevel optimization algorithm, can further elaborate its efficiency?
2. The algorithm for solving the bilevel optimization does not seem to obtain the optimal solution. Can its convergence be proved?

---

> ### Author Response · Authors · 2024-11-25
>
> Thank you for your helpful comments and questions.
>
> **Weakness 1:**
> In general, the experiments of the paper are relatively weak. Firstly, there is a limited comparison of the proposed algorithm with other notable approaches such as DeepProblog [1] and Semantic Loss [2]. Secondly, despite the authors conducting experiments on numerous datasets, these datasets appear to be predominantly toy and simplistic.
>
> **Response to Weakness 1:**
>
> Our work focuses primarily on examining whether different types of representational spaces—fuzzy rule-based spaces, graphical probabilistic model-based spaces, and distribution-based spaces—can be inter-mapped and unified within a continuous probabilistic framework. The core objective of our research lies in exploring and formalizing the transitions within continuous representations (**continuous <-> continuous**).
>
> In contrast, the scope of **DeepProbLog** [1] is fundamentally different, as it bridges discrete predicate logic to continuous representations by integrating neural network-based predicates (**discrete -> continuous**). Similarly, the scope of **Semantic Loss** [2] focuses on transitioning from continuous representations (e.g., neural network forward passes) to discrete label outputs for classification by employing constraint loss (**continuous -> discrete**).
>
> Given the divergence in scope and objectives, we chose not to include these approaches in our experimental comparisons, as they do not align directly with the core focus of our work.
>
>
> **Questions 1:**
> The author uses a bilevel optimization algorithm, can further elaborate its efficiency?
>
> **Response to Questions 1:**
>
> We acknowledge the reviewer's concerns regarding the efficiency of our bilevel optimization algorithm, particularly regarding the potential reliance on second-order information. In our implementation, we primarily employ approximate first-order optimization methods to address the bilevel optimization problem, ensuring computational efficiency. Specifically, we utilize widely adopted iterative gradient-based updating in [1] ("BOME! Bilevel Optimization Made Easy: A Simple First-Order Approach"), to implement approximate first-order updates effectively.
>
>
> **Questions 2:**
> The algorithm for solving the bilevel optimization does not seem to obtain the optimal solution. Can its convergence be proved?
>
> **Response to Question 2:**
>
> This work is not a theoretical paper focused on proving convergence, and we do not make any formal claims about convergence guarantees. Instead, we rely on mature Adam optimizer  to solve the subproblems with a reasonable level of accuracy.  Furthermore, the algorithm is executed for a sufficiently large number of iterations, and we monitor the stabilization of iterates and variables as a practical indicator of convergence.
>
> We respectfully request reconsideration of our score.

---

### Official Review · Reviewer_JjQC · 2024-11-04

**Soundness:** 3
**Presentation:** 2
**Contribution:** 3
**Rating:** 5
**Confidence:** 2

**Summary:**

This paper introduces a framework for neuro-symbolic learning called Mixed Multinomial Distribution-based NSL (MMD-NSL). The primary aim of the framework is to address two core challenges in NSL: managing long dependency chains and handling complex semantic categorization within knowledge graphs. This work propose a mixed multinomial logic semantic distribution to integrate both context-aware semantics and long-range dependencies, building upon traditional Markov Logic Networks.

The framework leverages a bilevel optimization strategy: the upper level, powered by transformer-based architectures, dynamically learns mixing coefficients analogous to attention mechanisms, while the lower level optimizes rule weights to capture context-specific dependencies.

**Strengths:**

The proposed MMD-NSL framework is novel in its combination of mixed multinomial distributions and bijective mapping between MLNs and continuous multinomial distributions. This dual approach uniquely extends MLNs to incorporate context-aware embeddings, bridging symbolic logic and continuous representations, which is both creative and forward-thinking.

The paper’s bilevel optimization strategy is technically sound and thoughtfully designed, enabling a transformer-based upper level to dynamically learn context-sensitive mixing coefficients while optimizing rule weights in the lower level.

**Weaknesses:**

1.Lack of Comparisons with More Diverse Baselines: The paper does not benchmark against a sufficiently broad spectrum of established baselines, which is essential for assessing relative performance comprehensively.

2.Distinction between Neuro-symbolic Learning and Causal and Temporal Reasoning: The differences between neuro-symbolic learning you've pointed out  and causal reasoning, temporal reasoning remain unclear. Clarifying how neuro-symbolic learning diverges from or intersects with causal and temporal reasoning approaches could enhance understanding of its unique capabilities and limitations.

3.Lack of Detailed Description for Training and Optimization Settings: The paper does not sufficiently detail the settings for training and optimization, which are crucial for replicating the results and understanding the model's performance under specific conditions.

**Questions:**

See the section on weakness. We will increase the score based on the answer to the question.

---

> ### Author Response · Authors · 2024-11-25
>
> Thank you for your helpful comments and questions.
>
> **Weakness 1:**
> Lack of Comparisons with More Diverse Baselines: The paper does not benchmark against a sufficiently broad spectrum of established baselines, which is essential for assessing relative performance comprehensively.
>
> **Response to Weakness 1:**
>
> The core principle of neuro-symbolic learning lies in mapping symbolic reasoning to various continuous spaces through differentiable methodologies tailored to specific settings and tasks. Our work more focuses on unifying and establishing bijections across different continuous mapping methods, providing a robust theoretical framework supported by strong mathematical underpinnings.
>
> As mentioned in Section 5.2, we carefully selected three representative approaches that are closely tied to our Theorems and Remarks. For instance, **Theorem 1** states: *"There exists a fuzzy logic semantic function $F_{\text{rule}}(z_j)$, in the form given in Eq.3, that establishes a bijective mapping from the MLN-based logic semantic distribution in Eq.1 to a multinomial distribution-based logic semantic distribution."*
>
> These three categories of approaches include:
> 1. **LTN** [1], which represents fuzzy logic-based continuous mapping and is used to evaluate Theorem 1.
> 2. **NMLN** [2], a neural network-based MLN continuous mapping method that can evaluate Remark 2.
> 3. **RNNLogic** [3] and **LogiRE** [4], which utilize multinomial distribution-based logic mapping and can evaluate Theorem 3.
>
> We believe that conducting theorem-driven ablation experiments is a more meaningful and rigorous strategy than simply stacking numerous empirical evaluation baselines, as it directly validates the foundational contributions of our framework.
>
> [1] Ivan Donadello, Luciano Serafini, and Artur S. d’Avila Garcez. Logic tensor networks for semantic image interpretation. In Carles Sierra, editor, IJCAI, 2017.
> [2] Giuseppe Marra and Ondrej Kuželka. Neural markov logic networks. CoRR, abs/1905.13462, 2019.
> [3] Qu M, Chen J, Xhonneux LP, Bengio Y, Tang J. RNNLogic: Learning Logic Rules for Reasoning on Knowledge Graphs. InInternational Conference on Learning Representations.
> [4] Ru, Dongyu, et al. "Learning Logic Rules for Document-Level Relation Extraction." Proceedings of the 2021 Conference on Empirical Methods in Natural Language Processing. 2021.
>
> **Weakness 2:**
> Distinction between Neuro-symbolic Learning and Causal and Temporal Reasoning: The differences between neuro-symbolic learning you've pointed out and causal reasoning, temporal reasoning remain unclear. Clarifying how neuro-symbolic learning diverges from or intersects with causal and temporal reasoning approaches could enhance understanding of its unique capabilities and limitations.
>
> **Response to Weakness 2:**
> We have revised the introduction and added a framework overview to clearly define the reasoning tasks our Neuro-symbolic Learning (NSL) method is designed to address. These revisions aim to enhance the reader's understanding of the scope and applicability of our approach.
>
> It is important to emphasize that causal reasoning and temporal reasoning are fundamentally distinct from NSL and fall outside the scope of related work for this paper. Instead, our work is closely aligned with logical reasoning, which we have elaborated on in the related work section, with key updates highlighted in red.
>
> Causal reasoning involves identifying and analyzing cause-and-effect relationships between events or variables. For example, it is often applied to identify risk factors for diseases by determining what causes specific health conditions.
>
> Temporal reasoning, in contrast, deals with the timing, sequence, and dependencies of events. It is widely applied in scenarios such as planning, scheduling, and analyzing temporal knowledge graphs.
>
> Neuro-symbolic learning, on the other hand, focuses on representing and reasoning with logical symbolic structures within a continuous, probabilistic, and differentiable framework.
>
> The unique contribution of our work lies in going beyond traditional logical reasoning by incorporating contextual information into a distributional framework. This allows our NSL method to handle more complex scenarios where reasoning depends not only on long dependency chains but also on diverse and dynamic contextual information.
>
> **Weakness 3:**
> Lack of Detailed Description for Training and Optimization Settings: The paper does not sufficiently detail the settings for training and optimization, which are crucial for replicating the results and understanding the model's performance under specific conditions.
>
> **Response to Weakness 3:**
> We have added a model optimization setup with section 4.2 with red mark to describe the training and optimization settings.

---

> > ### Comment · Reviewer_JjQC · 2024-11-27
> >
> > Dear authors, thanks for your answer and clarifications.
> >
> > The updates made to the methods setup section are an improvement. However, it would be even more valuable to explore the key distinctions between your approach and the systems discussed. I still believe that conducting some experimental comparisons with these systems is also essential.

---

### Meta-Review · Area_Chair_JR6Y · 2024-12-19

**Metareview:**

This paper introduces the Mixed Multinomial Distribution-based Neuro-Symbolic Learning (MMD-NSL) framework, designed to tackle long dependency chains and complex semantic categorization in knowledge graphs. It extends traditional Markov Logic Networks (MLNs) by integrating context-aware semantic embeddings and a continuous mixed multinomial logic semantic distribution. The framework employs a bilevel optimization strategy, leveraging transformer-based architectures for dynamic mixing coefficients and rule weight optimization. The integration of mixed multinomial distributions and bijective mapping between MLNs and continuous multinomial distributions is innovative and extends classical NSL. However, the paper lacks comprehensive comparisons with a broader range of established baselines, limiting the assessment of its relative performance. The experiments focus on simplistic datasets, and the training/optimization settings are insufficiently detailed, making replication challenging. Therefore, I recommend the rejection of this paper. The work requires further refinement before submission to a future venue.

**Additional Comments On Reviewer Discussion:**

After the rebuttal, the reviewers still maintain that the experimental section of the current version is insufficient. Moreover, no reviewer defends that the strengths of the paper outweigh its weaknesses.

---

### Decision · Program_Chairs · 2025-01-22

Reject